# Modeling opening price spread of Shanghai Composite Index based on ARIMA-GRU/LSTM hybrid model

**Yuancheng Si**[1]*, **Saralees Nadarajah**[2], **Zongxin Zhang**[1], **Chunmin Xu**[3]

1 School of Economics, Fudan University, ShangHai, PR China, 2 Department of Mathematics, University of Manchester, Manchester, United Kingdom, 3 Jiangxi Medical College, Nanchang University, Nanchang, China

☯ These authors contributed equally to this work.

* siyuancheng2054@163.com

**Data Availability Statement:** All relevant data are within the paper and its Supporting information files.

**Funding:** The author(s) received no specific funding for this work.;

## Abstract

In the dynamic landscape of financial markets, accurate forecasting of stock indices remains a pivotal yet challenging task, essential for investors and policymakers alike. This study is motivated by the need to enhance the precision of predicting the Shanghai Composite Index's opening price spread, a critical measure reflecting market volatility and investor sentiment. Traditional time series models like ARIMA have shown limitations in capturing the complex, nonlinear patterns inherent in stock price movements, prompting the exploration of advanced methodologies. The aim of this research is to bridge the gap in forecasting accuracy by developing a hybrid model that integrates the strengths of ARIMA with deep learning techniques, specifically Long Short-Term Memory (LSTM) and Gated Recurrent Unit (GRU) networks. This novel approach leverages the ARIMA model's proficiency in linear trend analysis and the deep learning models' capability in modeling nonlinear dependencies, aiming to provide a comprehensive tool for market prediction. Utilizing a comprehensive dataset covering the period from December 20, 1990, to June 2, 2023, the study develops and assesses the efficacy of ARIMA, LSTM, GRU, ARIMA-LSTM, and ARIMA-GRU models in forecasting the Shanghai Composite Index's opening price spread. The evaluation of these models is based on key statistical metrics, including Mean Squared Error (MSE) and Mean Absolute Error (MAE), to gauge their predictive accuracy. The findings indicate that the hybrid models, ARIMA-LSTM and ARIMA-GRU, perform better in forecasting the opening price spread of the Shanghai Composite Index than their standalone counterparts. This outcome suggests that combining traditional statistical methods with advanced deep learning algorithms can enhance stock market prediction. The research contributes to the field by providing evidence of the potential benefits of integrating different modeling approaches for financial forecasting, offering insights that could inform investment strategies and financial decision-making.

**Competing interests:** The authors have declared that no competing interests exist.

# 1 Introduction

In the stock market, an important phenomenon is price difference, commonly referred to as the opening price difference or morning price difference [1–3]. These terms describe the situation where there is a difference between the opening price of a security or index and its previous day's closing price. This phenomenon is observed in various financial markets, including stock and commodity markets. When we talk about price difference, it can be further divided into positive price difference and negative price difference. A positive price difference indicates that the opening price is higher than the previous day's closing price, which may be due to positive information received by the market after the close, leading buyers to be willing to pay a higher price at market open. On the other hand, a negative price difference indicates a lower opening price, which may be attributed to negative information received by the market after the close, causing sellers to be willing to sell at a lower price at market open [3, 4].

In summary, the phenomenon of price differences reflects an important characteristic of financial markets, namely, price volatility and uncertainty. The occurrence of price differences provides a unique perspective that helps us understand and explain the volatility of stock prices. In previous studies, the phenomenon of price differences has received extensive attention. Many researchers [3, 5, 6] have conducted in-depth research on price differences in different financial markets, attempting to understand the causes and impacts of this phenomenon and establish daily price difference models to validate relevant economic hypotheses [3, 7]. Among them, some studies have found that market information efficiency is an important factor affecting price differences [8]. In markets with high information efficiency, price differences may be less common because all information has been fully absorbed by market participants and reflected in prices. However, in markets with low information efficiency, price differences may be more frequent as market participants may not immediately access and react to all information. Some empirical analyses of the US stock market have also shown that adjustments to bad news are faster than those to good news [6, 9]. After a positive opening price difference, stock prices tend to have stronger upward momentum, providing opportunities for profitable trading. Overall, price differences are an important phenomenon in the stock market, and a deep understanding and research of this phenomenon are significant for revealing the volatility of stock prices, understanding and predicting market dynamics, and making investment decisions. The remainder of this document unfolds as follows: Section 2 reviews relevant literature, providing a theoretical foundation and contextual background. Section 3 presents the data collection methodology, including the sources, variables, and descriptive statistics of the dataset utilized. Section 4 delves into the rationale behind using overnight returns as a proxy for market sentiment. Section 5 explores the relationship between trading volume ratios and sentiment, offering empirical evidence. Section 6 subjects the study's findings to a series of robustness checks to verify their validity. Finally, Section 7 concludes the paper, summarizing the key insights and suggesting directions for future research.

# 2 Literature review

From a theoretical perspective, according to the Efficient Market Hypothesis (EMH) [10, 11], in an efficient market, asset prices will fully and promptly reflect all available information. Thus, there is a close relationship between the Efficient Market Hypothesis and stock index price differences. In a perfectly efficient market, investors cannot obtain excess returns through any means, as all information is fully absorbed by the market and reflected in prices. However, the occurrence of opening price differences suggests a certain degree of market inefficiency. Specifically, if there is a difference between the closing price of the previous trading day and the opening price of the current day, it may indicate that the market did not fully

incorporate all information into the closing price of the previous trading day or that there is a delay in the market's reaction to new information [1, 12]. Although stock index price differences imply possible market inefficiency, it does not mean that investors can easily obtain excess returns. In practice, investors need to consider many other factors, including transaction costs, market volatility, and other factors that may affect investment decisions. Moreover, even if there is a certain degree of market inefficiency, the Efficient Market Hypothesis reminds us that this inefficiency may quickly disappear as market participants react.

From a practical perspective, [3] study rigorously examines the Ukrainian stock market, specifically analyzing the UX index from 2009 to 2018, to discern patterns and anomalies related to price gaps. Through comprehensive statistical and regression analyses, it finds no significant evidence of seasonality or abnormal behavior post-gaps, aligning with the Efficient Market Hypothesis, except for a momentum effect on days with negative gaps, suggesting a profitable trading strategy that contradicts market efficiency. Similarly, [13] undertakes an extensive analysis of price gaps across stock, FOREX, and commodity markets from 2000 to 2015, employing various statistical tests to explore six hypotheses regarding market efficiency. It concludes that while most market behaviors align with efficiency, FOREX markets exhibit an anomaly that allows for the generation of abnormal profits through a specific trading strategy, highlighting a distinct deviation from market efficiency in the FOREX sector. Adding to the complexity of market efficiency, recent studies have delved into the nuanced dynamics of stock market [5, 14, 15], revealing intriguing exceptions to the momentum effect. A novel examination into this market demonstrates that intraday and overnight returns significantly influence future stock returns in differing manners. Investors show a tendency to underreact to intraday information, while overreacting to overnight information, leading to the formulation of intraday momentum and overnight momentum strategies. This dichotomy not only challenges the traditional understanding of market reactions but also illustrates the persistence of profitability, showcasing its resilience against momentum crashes. Furthermore, the relationship between overnight returns and investor sentiment on the Taiwan Stock Exchange (TWSE) has been reassessed [15], corroborating the findings by Aboody et al. that overnight returns reflect investor sentiment. This study extends the understanding by highlighting how trading activities by different investor types amplify the patterns of overnight returns, with a significant role played by retail trading volume. It elucidates that overnight returns contribute to both short-term persistence and long-term return reversals, driven by investor sentiment. These insights not only validate the use of overnight returns as a measure of investor sentiment in the TWSE but also suggest the influence of market structure and investor behaviors as critical determinants in non-US markets [16]. These recent findings enrich the discourse on market efficiency by illustrating how specific market mechanisms and investor behaviors can lead to anomalies that both challenge and complement the Efficient Market Hypothesis. They underscore the importance of considering intraday and overnight information separately in analyzing market dynamics and formulating trading strategies. To bridge the insights from specific market behaviors and anomalies highlighted in [3, 13] with the broader considerations of market dynamics, it's imperative to understand the underlying factors contributing to price differences. Factors contributing to price differences can be summarized as follows:

• Announcement of macroeconomic information [1, 12]: Important macroeconomic information of a country or other countries may be announced after the trading market closes, such as adjustments in monetary policy or the release of GDP growth data. These announcements can influence investors' expectations and result in differences between the opening price and the previous day's closing price.

- Leakage of internal corporate information [6, 12]: Significant leaks of corporate information, such as financial reports or major decisions, can have an impact on stock prices. Particularly, when this information is disclosed after the trading market closes or leaked before the market opens, it often leads to significant differences between the opening price and the previous day's closing price.

- Changes in market microstructure information [17]: Changes in market microstructure information, including trading volume and frequency, can affect stock prices. When these changes occur, they may result in differences between the opening price and the previous day's closing price.

- Changes in non-economic market sentiment [7, 18]: Market sentiment is an important factor influencing stock prices. Investor panic or excessive optimism can lead to abnormal price fluctuations. When market sentiment changes significantly after
the trading market closes due to non-economic factors such as geopolitical events or natural disasters, it often results in significant differences between the opening price and the previous day's closing price for specific stocks or sectors.

- Liquidity shocks [12]: Liquidity shocks refer to sudden changes in market liquidity, such as a concentration of buy or sell orders at the moment the market opens or a sudden inflow or outflow of funds. Such liquidity shocks can lead to significant differences between the opening price and the previous day's closing price.

Despite the in-depth exploration of price differences in previous research from different perspectives, considering the complexity and diversity of their influences, as well as the continuously changing market environment and investor behavior, further deepening and expanding the research on this phenomenon is still needed. In the context of price difference modeling, we often encounter several core problems and demands:

1. Simplicity and effectiveness: Autoregressive models are simple and applicable in many situations. They only require knowledge of the historical data of a single variable to make predictions. This makes the model easy to understand and interpret, especially in the financial field, where stock prices are often believed to be influenced by past prices [19, 20].

2. Autocorrelation: In many financial time series data, there is autocorrelation between the data at a certain time point and its historical data, meaning that the current value may depend on past values. This dependency is the core assumption of autoregressive models, which often holds true in many financial time series.

3. Noise in high-frequency data: When using high-frequency financial data (such as data per minute or per second), the data may contain a large amount of noise. Introducing more covariates may make the model too complex and introduce the risk of overfitting. On the other hand, autoregressive models, due to their simplicity, can better handle this noise [21].

4. Data availability: In practical applications, we may not have access to data for all variables that may influence stock prices, or the cost of obtaining such data may be high.

Meanwhile, obtaining comprehensive datasets that cover all factors influencing stock prices presents significant challenges, including data availability and associated costs. High-quality, extensive financial data often requires substantial investment, limiting accessibility for individual researchers and smaller entities. Moreover, the complexity of accurately analyzing and integrating this data to reflect stock price dynamics poses additional hurdles. The integration of covariates from diverse frequencies—daily to quarterly—into a coherent predictive model

requires advanced methodologies to maintain data integrity and avoid analytical bias. Exploring the impact of such covariates on the opening price spread offers a promising research direction, potentially uncovering the underlying mechanisms of stock market movements. This exploration could lead to more accurate stock price predictions and informed trading strategies, necessitating access to detailed, high-frequency data and the application of sophisticated statistical and machine learning techniques. Given these considerations, This study adopts a hybrid approach, combining autoregressive modeling with deep learning, to address the complexities of financial time series analysis effectively. This strategy is designed to harness the strengths of both methodologies, enhancing the ability to predict stock price variations despite the challenges posed by integrating diverse covariates. The novelty of this study is reflected in the following aspects:

1. Comprehensive Data Analysis of Shanghai Composite Index's Opening Price Spread: This study represents one of the first comprehensive analyses focusing specifically on the opening price spread rate of the Shanghai Composite Index. We delve deep into the historical data, meticulously analyzing patterns and trends over an extensive period. This approach offers a unique perspective on how the opening price spread behaves in one of the world's largest financial markets, providing new insights into market dynamics.

2. Hybrid Modeling Approach for Enhanced Forecast Accuracy: We adopt a novel hybrid modeling strategy, combining the strengths of autoregressive models and deep learning techniques. This approach effectively addresses the limitations of traditional time series models in capturing complex, nonlinear relationships inherent in stock prices. By integrating ARIMA with advanced deep learning models like LSTM and GRU, we significantly improve the accuracy of forecasting the opening price spread. This methodological innovation marks a substantial advancement over traditional forecasting models.

3. Focus on Practical Implications and Application: This study goes beyond theoretical exploration and actively addresses practical implications. We provide insights that are directly applicable for investors, traders, and financial analysts. By improving the accuracy of forecasting models, we offer tools that can aid in more informed decision-making processes in the realm of investment and trading strategies. This practical focus ensures that the research is not only academically relevant but also of tangible value to those operating in financial markets. These novel elements of this study contribute to the existing body of knowledge in financial time series analysis and forecasting. They demonstrate the effectiveness of this approach in addressing the unique challenges presented by the financial data and highlight the potential for future research in this area, particularly in exploring the impact of various covariates on stock market dynamics.

## 3 Date and methods

### 3.1 Data

We downloaded the dataset of the Shanghai Stock Exchange Composite Index from December 20, 1990, to June 2, 2023, comprising a total of 7,927 trading days (excluding non-trading days such as statutory holidays) from Sina Finance(https://finance.sina.com.cn/stock/). The dataset consists of 9 feature columns, where each sample represents a day of stock market trading data. The following are the feature columns in the dataset and their meanings:

- 'date': Date

- 'pre_close': Previous day's closing price

- 'open': Opening price on the current day

- 'high': Highest price on the current day

- 'low': Lowest price on the current day

- 'close': Closing price on the current day

- 'changeinprice': Price change

- 'changeinrate': Price change rate

- 'diffrate': Opening price difference rate

In this dataset, the focus is on the 'diffrate' (opening price difference rate) column. 'diffrate' represents the percentage difference between the opening price of the day and the previous day's closing price. By calculating the difference between the daily opening price and the previous day's closing price, dividing it by the previous day's closing price, and multiplying by 100, we obtain the opening price difference rate as a percentage.

$$diffrate = \left( \frac{open - pre\_close}{pre\_close} \right) \times 100\% \qquad (1)$$

## 3.2 Interpretation of opening price difference rate

The opening price difference rate provides information about the relative change between the daily opening price and the previous day's closing price. A positive value indicates that the opening price is higher than the previous day's closing price, while a negative value indicates that the opening price is lower. This indicator helps us understand market volatility, trends, and the magnitude of price changes. It reflects market sentiment: the opening price difference rate can be considered as one of the sentiment indicators of market participants. A positive value suggests that market participants have positive expectations for the stock or index, leading them to be willing to buy at a higher price. On the other hand, a negative value may reflect market participants' concerns or cautious sentiment. Additionally, the opening price difference rate can also reflect market supply and demand dynamics. If the opening price is higher than the previous day's closing price, it may indicate higher market demand with more buyers, resulting in price increases. Conversely, if the opening price is lower, it may indicate higher market supply with more sellers, leading to price declines. Analyzing the opening price difference rate helps explore market trends and volatility. For example, when the absolute value of the opening price difference rate is large, it may indicate significant market fluctuations or unexpected events that can have a significant impact on the stock market. This is valuable for investors, traders, and analysts in predicting market trends and formulating appropriate investment strategies.

## 3.3 Exploratory data analysis and unit root test

We conducted exploratory data analysis on the 'diffrate' (opening price difference rate) using statistical software to understand its economic and statistical significance. The 'diffrate' represents the percentage difference between the daily opening price and the previous day's closing price. The data span from December 20, 1990, to June 2, 2023, comprising a total of 7,927 samples.

Descriptive statistics of the 'diffrate' are presented in Table 1. According to the statistics, the 'diffrate' has a mean of 0.008907 and a standard deviation of 1.572218. This indicates that, on

**Table 1. Group counts of 'diffrate'.**

| Group | Count | Group | Count | Group | Count | Group | Count |
|---|---|---|---|---|---|---|---|
| [-10.0, -9.5) | 1 | [-9.5, -9.0) | 2 | [-9.0, -8.5) | 2 | [-8.5, -8.0) | 2 |
| [-8.0, -7.5) | 2 | [-7.5, -7.0) | 2 | [-7.0, -6.5) | 3 | [-6.5, -6.0) | 3 |
| [-6.0, -5.5) | 5 | [-5.5, -5.0) | 1 | [-5.0, -4.5) | 5 | [-4.5, -4.0) | 9 |
| [-4.0, -3.5) | 14 | [-3.5, -3.0) | 20 | [-3.0, -2.5) | 21 | [-2.5, -2.0) | 48 |
| [-2.0, -1.5) | 92 | [-1.5, -1.0) | 186 | [-1.0, -0.5) | 528 | [-0.5, 0.0) | 3099 |
| [0.0, 0.5) | 2969 | [0.5, 1.0) | 549 | [1.0, 1.5) | 157 | [1.5, 2.0) | 76 |
| [2.0, 2.5) | 37 | [2.5, 3.0) | 24 | [3.0, 3.5) | 15 | [3.5, 4.0) | 8 |
| [4.0, 4.5) | 6 | [4.5, 5.0) | 6 | [5.0, 5.5) | 4 | [5.5, 6.0) | 4 |
| [6.0, 6.5) | 5 | [6.5, 7.0) | 4 | [7.0, 7.5) | 2 | [7.5, 8.0) | 3 |
| [8.0, 8.5) | 2 | [9.0, 9.5) | 2 | [9.5, 10.0) | 2 | [10.0, 10.5) | 1 |

average, the daily opening price has a relatively small difference compared to the previous day's closing price, but it also exhibits significant volatility. The minimum value is -21.822000 (occurred on August 12, 1992, during a period when the market experienced a rare and drastic decline from its peak of 1429.01 points in May 1992, dropping to around 600 points in September before rebounding to around 700 points), and the maximum value is 104.269000 (occurred on May 21, 1992, when the Shanghai Stock Exchange lifted the price limit on the only 15 listed stocks, triggering a market surge. Without the limit-up rule, the Shanghai market rose 105% in a single day. Note: In the histogram, we excluded the corresponding extreme values). These values demonstrate the wide range of variations in the 'diffrate' throughout the study period.

The histogram of the 'diffrate' is shown in Fig 1 to visualize its distribution. The histogram illustrates the frequency distribution of the 'diffrate' within different intervals. From the

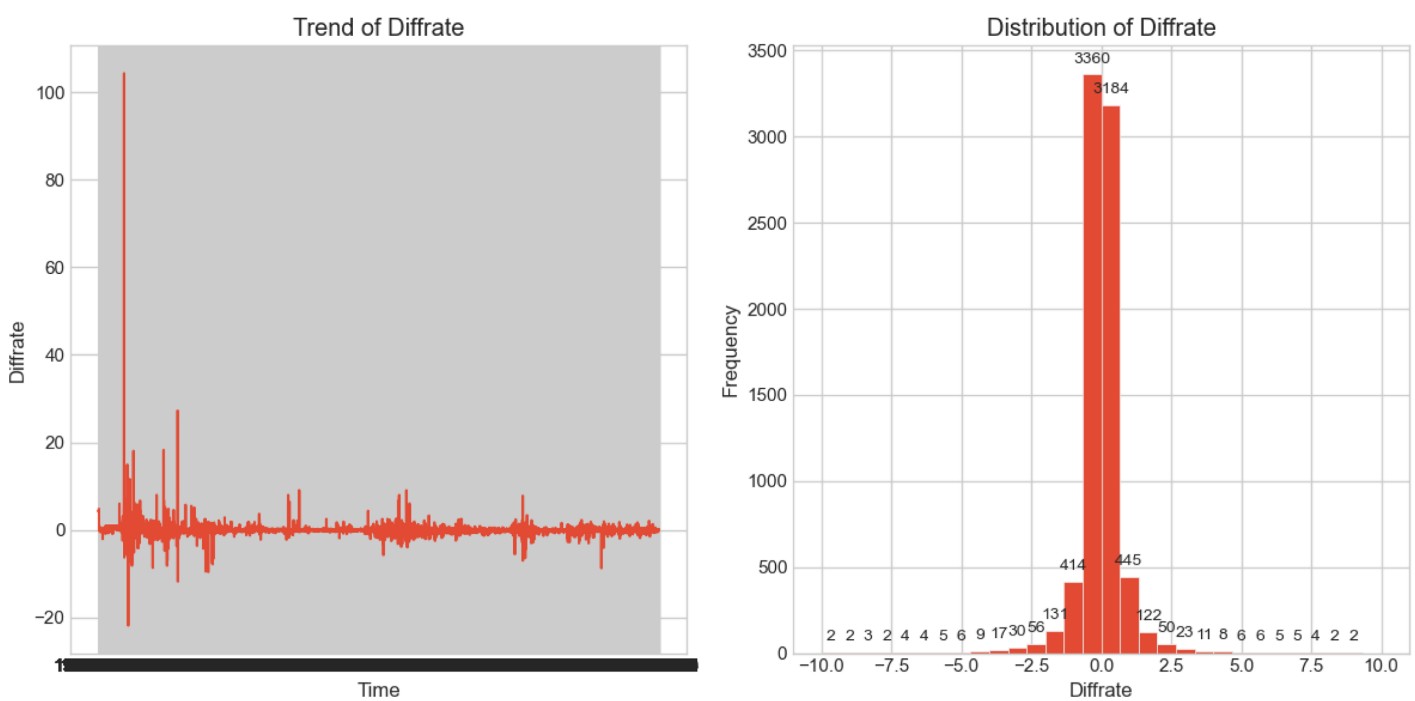

**Fig 1. Histogram and time series analysis of diffrate.**

**Table 2. Statistics of 'diffrate'.**

|  | count | mean | std | min | 25% | 50% | 75% | max |
|---|---|---|---|---|---|---|---|---|
| diffrate | 7927 | 0.009 | 1.572 | -21.822 | -0.215 | -0.007 | 0.198 | 104.269 |

histogram, we observe that the 'diffrate' is mainly concentrated around smaller values, but there are also some extreme values. This indicates that, during the study period, the opening price's relative change compared to the previous day's closing price was mostly small, but occasionally, there were significant changes. We also provide binned counts of the non-zero intervals of the entire 'diffrate' with a bin size of 0.5 as statistics of 'diffrate', as shown in Tables 1 and 2.

By plotting the line chart of the 'diffrate' over time as shown in Fig 1, we further explore its trends and volatility. From the line chart, we can observe that the 'diffrate' exhibits significant fluctuations at different time points, displaying clear upward and downward trends. This further confirms the importance of the 'diffrate' as a key indicator that can help us predict market trends and formulate corresponding investment strategies.

In order to conduct time series modeling on 'diffrate', we first performed a unit root test. The unit root test is a commonly used method in time series analysis to determine whether a series has a random drift or is non-stationary. For this study, the purpose of the unit root test is to determine whether 'diffrate' needs to be differenced to become a stationary time series, which is suitable for ARIMA modeling. In the unit root test, we used the Augmented Dickey-Fuller Test (ADF test), which is one of the commonly used unit root test methods. The null hypothesis of the ADF test is that the series has a unit root (non-stationary), while the alternative hypothesis is that the series is stationary. If the test result rejects the null hypothesis, indicating significant statistical evidence that the series is stationary, we can proceed with further modeling analysis using the differenced series. We used the 'adfuller' function from the 'statsmodels' library to perform the ADF test, and the results are as follows:

The ADF statistic is -17.5854 and the p-value is 0.0000. At a significance level of 0.05, the p-value is less than the significance level, indicating that we can reject the null hypothesis. This means that 'diffrate' is a stationary time series rather than having a unit root (non-stationary). Additionally, we compared the ADF statistic with critical values. For a 1% significance level, the critical value is -3.4312; for a 5% significance level, the critical value is -2.8619; and for a 10% significance level, the critical value is -2.5670. Since the ADF statistic is much smaller than the critical values, it further supports the conclusion that 'diffrate' is a stationary time series. Based on these results, we can confirm that 'diffrate' is a stationary time series. This means that we can directly use 'diffrate' as the input data in the modeling process without differencing. Therefore, we can proceed with ARIMA modeling using 'diffrate' to predict market trends and formulate related investment strategies.

In summary, 'diffrate' has significant economic and statistical significance. It reflects market sentiment and supply-demand dynamics, helping us predict market trends and volatility. Through exploratory data analysis, we gained a deeper understanding of the distribution characteristics and trends of 'diffrate', laying the foundation for further research and analysis.

## 3.4 Modeling and forecasting process

According to the time order, we sorted the entire dataset in ascending order, from the earliest to the most recent. This was done to ensure that the model predicts and evaluates based on the temporal continuity during the training and testing process.

| day1 | day2 | day3 | day4 | day5 | day6 | day7 | day8 | day9 | day10 | day11 | day12 | day13 | day14 | day15 |
|------|------|------|------|------|------|------|------|------|-------|-------|-------|-------|-------|-------|
| | | | | | train set | | | | | prediction | | | | |
| | | | | | | train set | | | | | prediction | | | |
| | | | | | | | train set | | | | | prediction | | |
| | | | | | | | | train set | | | | | prediction | |
| | | | | | | | | | train set | | | | | prediction |

**Fig 2. Illustration of rolling forecast methodology.**

Next, we divided the data into two parts: the first 80% of the data was used as the training set for the entire modeling process, and the remaining 20% was set aside as the validation set to evaluate the predictive performance of the model. To accurately assess the model's predictive ability, we employed a rolling forecasting approach for validation. Specifically, we trained the model using a fixed window of historical data, then predicted the value for the next time point and compared it with the actual observed value. This approach simulates the forecasting scenario in practical applications and provides a better assessment of the model's generalization ability on future data. See Fig 2 for a visualization of the process.

In this study, we will compare the performance of five models, including ARIMA, LSTM, GRU, and hybrid models ARIMA-LSTM and ARIMA-GRU. These models will be evaluated through rolling forecasts on the test set, and their predictive performance will be measured using evaluation metrics such as root mean square error, mean absolute error, prediction accuracy, etc. We will analyze and compare the forecasting results of each model to determine which model performs best in predicting the "opening price difference".

## 4 Model composition

### 4.1 ARIMA model

The Autoregressive Integrated Moving Average Model(ARIMA) model, is a statistical model used for analyzing and forecasting time series data [21]. The ARIMA model consists of three main components: Autoregressive (AR) component, Integrated (I) component, and Moving Average (MA) component. Specifically, the AR component assumes that the current value depends on past p observations, where p is a parameter to be determined. The MA component assumes that the current value depends on past q error terms, where q is also a parameter to be determined. The I component is responsible for transforming non-stationary series into stationary ones. The general form of the ARIMA model can be written as $ARIMA(p, d, q)$, where p is the order of the autoregressive component, d is the degree of differencing, and q is the order of the moving average component.

The mathematical expression of the ARIMA model is as follows:

$$\left(1 - \sum_{i=1}^{p} \phi_i L^i\right)(1 - L)^d X_t = \left(1 + \sum_{i=1}^{q} \theta_i L^i\right)\varepsilon_t \tag{2}$$

where $\phi_i$ represents the autoregressive coefficients, $\theta_i$ represents the moving average coefficients, $L$ is the lag operator, $X_t$ is the time series, $\varepsilon_t$ is the error term, and $d$ is the degree of differencing.

Fitting the time series data with ARIMA(p, d, q) means utilizing a combination of different orders of AR, MA, and ARMA to capture various patterns and information in the time series, thus achieving effective time series forecasting. ARIMA models are commonly used to fit non-stationary financial time series ARIMA is often chosen as a baseline model in time series

forecasting due to its robustness and simplicity. It is particularly useful for establishing a benchmark for comparison with more complex models. Below we detail the reasons for selecting ARIMA as the baseline model in this study:

- **Simplicity and Interpretability:** ARIMA models are straightforward and their parameters have clear interpretations, which is valuable for initial analysis and benchmarking.

- **Maturity and Stability:** The ARIMA model is a well-established method in time series analysis, providing reliable and consistent benchmarks.

- **Less Data Requirement:** ARIMA can perform well even with smaller datasets, making it suitable for situations where data availability is limited.

- **Benchmarking:** It provides a standard against which the performance of more complex models can be compared. This is crucial in assessing whether the additional complexity of newer models translates into better performance.

- **Theoretical Foundations:** ARIMA is grounded in statistical theory, which helps in understanding the underlying processes in the time series data.

Given these reasons, ARIMA serves as the baseline model in this analysis, against which the performance of ARIMA-LSTM, LSTM, GRU, and ARIMA-GRU is evaluated in section 8.

## 4.2 Deep learning models

**4.2.1 Long Short-Term Memory (LSTM) model.** Long Short-Term Memory (LSTM) is a special type of Recurrent Neural Network (RNN) proposed by Hochreiter and Schmidhuber in 1997 [22]. It addresses the issue of handling long-term dependencies in traditional RNNs when dealing with long sequences. LSTM is particularly suitable for modeling and predicting important events in time series with relatively long intervals and delays. Many researchers have applied LSTM models to traditional time series forecasting and achieved promising results [19, 23].

LSTM introduces a structure called a "memory cell" to store and access long-term information. Each memory cell is controlled by an input gate, a forget gate, and an output gate. The control signals of these gates are computed using sigmoid functions, which produce values between 0 and 1, indicating the degree of gate opening.

Specifically, the computation process of LSTM can be expressed with the following mathematical equations:

- Input Gate: $i_t = \sigma(W_{ii}x_t + b_{ii} + W_{hi}h_{t-1} + b_{hi})$

- Forget Gate: $f_t = \sigma(W_{if}x_t + b_{if} + W_{hf}h_{t-1} + b_{hf})$

- Output Gate: $o_t = \sigma(W_{io}x_t + b_{io} + W_{ho}h_{t-1} + b_{ho})$

- New Cell State: $g_t = \tanh(W_{ig}x_t + b_{ig} + W_{hg}h_{t-1} + b_{hg})$

- Updated Cell State: $c_t = f_t{}^*c_{t-1} + i_t{}^*g_t$

- Final Hidden State: $h_t = o_t{}^*\tanh(c_t)$

In these equations:$h_t$ represents the hidden state at time step $t$, $x_t$ represents the input at time step $t$, $c_t$ represents the cell state at time step $t$, $W$ and $b$ denote weight matrices and bias vectors, with subscripts indicating the input (i), hidden (h), and gate types (i, f, o, g), $\sigma$ represents the sigmoid function, tanh represents the hyperbolic tangent function, $^*$ denotes element-wise multiplication (Hadamard product), $i_t, f_t, o_t,$ and $g_t$ represent the candidate values for the input gate, forget gate, output gate, and cell state, respectively.

The development of LSTM was aimed at addressing the challenges that traditional RNNs face when dealing with long sequential data. Traditional RNNs struggle to capture long-term dependencies in sequences, making it difficult to capture dependencies with long gaps. LSTM effectively addresses this issue by introducing memory cells and gate mechanisms.

The memory cell in LSTM allows for the storage and access of long-term information. It maintains a hidden state that can be updated selectively using gate mechanisms. The gate mechanisms include the input gate, forget gate, and output gate, which control the flow of information in and out of the memory cell. The input gate determines how much new information should be stored in the memory cell, while the forget gate determines how much old information should be discarded. The output gate controls the information that is outputted from the memory cell.

By incorporating memory cells and gate mechanisms, LSTM can effectively capture long-term dependencies in sequential data. It can learn and remember information over extended periods, allowing it to model and predict sequences with long gaps between relevant events. This makes LSTM a powerful tool for tasks such as language modeling, machine translation, speech recognition, and time series analysis.

**4.2.2 Gated Recurrent Unit (GRU) model.** Gated Recurrent Unit (GRU) is a variation of Recurrent Neural Networks (RNNs) used for processing sequential data such as time series, speech, and text [22, 23]. It addresses the issue of vanishing or exploding gradients that can occur in RNNs when dealing with long sequences. These issues arise from the fact that gradients may decrease (vanish) or increase (explode) with the increase in time steps during the backpropagation process, making it difficult for the network to learn and remember long-term information. GRU is designed to effectively store and process information over long sequences. It is a simplified version of LSTM, reducing the number of gates and merging the cell state and hidden state.

A key feature of GRU is its update gate and reset gate. The update gate determines the extent to which the old hidden state should be retained when updating the hidden state, while the reset gate determines the extent to which the old hidden state should be discarded when computing the new candidate hidden state. These gates allow GRU to capture long-term dependencies in time series effectively. See Fig 3 for the graphical representation.

The mathematical expressions for GRU are as follows [22]:

- Reset Gate: $r_t = \sigma(W_r x_t + U_r h_{t-1} + b_r)$

- Update Gate: $z_t = \sigma(W_z x_t + U_z h_{t-1} + b_z)$

- Candidate Hidden State: $\tilde{h}_t = \phi(W_h x_t + U_h(r_t \odot h_{t-1}) + b_h)$

- Final Hidden State: $h_t = (1 - z_t) \odot h_{t-1} + z_t \odot \tilde{h}_t$

where the operator $\odot$ represents the element-wise multiplication (Hadamard product). In these equations: $x_t$ is the input vector, $h_t$ is the output vector, $\tilde{h}_t$ is the candidate activation vector, $z_t$ is the update gate vector, $r_t$ is the reset gate vector, $W$, $U$, and $b$ are the parameter matrices and vectors.

GRU has the following advantages, making it more suitable for modeling and analyzing sequence data in certain scenarios compared to Long Short-Term Memory (LSTM) networks:

- Simplified structure: GRU has two gates (reset gate and update gate) compared to LSTM's three gates (forget gate, input gate, and output gate). With fewer parameters, GRU is typically easier to train and faster than LSTM.

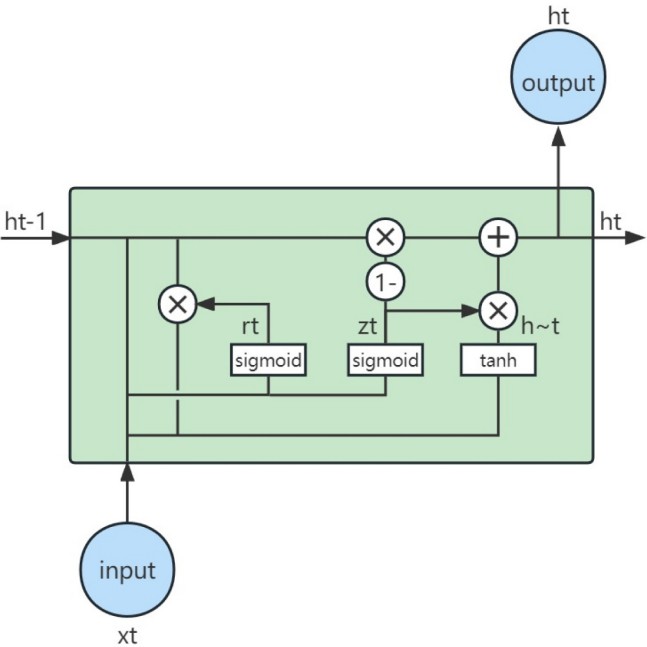

**Fig 3. Capture structure of GRU model.**

- Computational efficiency: Due to fewer parameters, GRU is computationally efficient, providing substantial benefits when dealing with large-scale data.

- Suitable for short and low-complexity sequences: Studies have shown that GRU performs better than LSTM networks on low-complexity sequences.

## 5 Algorithm procedure for hybrid model

In many time series models, both linear and nonlinear relationships are considered. The ARIMA model performs well in capturing linear relationships in time series data, but it has limitations in modeling nonlinear relationships. The LSTM model, on the other hand, is capable of modeling both linear and nonlinear relationships, but its performance may vary when applied to different datasets. To achieve optimal prediction results, researchers have adopted hybrid models that leverage the principles of separately modeling the linear and nonlinear components of time series. These models have achieved great success in time series analysis and prediction by utilizing various deep learning algorithms to achieve better estimation performance than constructive learning algorithms. Additionally, these models belong to supervised learning algorithms, which can be used for training and prediction to achieve superior results. The process of predicting price differentials in financial time series using hybrid models is a complex and meticulous task that requires the application of various statistical and financial methods and models. The specific steps involved are as follows Fig 4.:

1. Perform descriptive statistical analysis on the high-frequency financial time series to observe its distributional characteristics. This step serves as the foundation of the prediction process, as understanding the basic properties of the data allows for a better grasp of its structure and patterns. To meet the requirements of the model, data preprocessing is

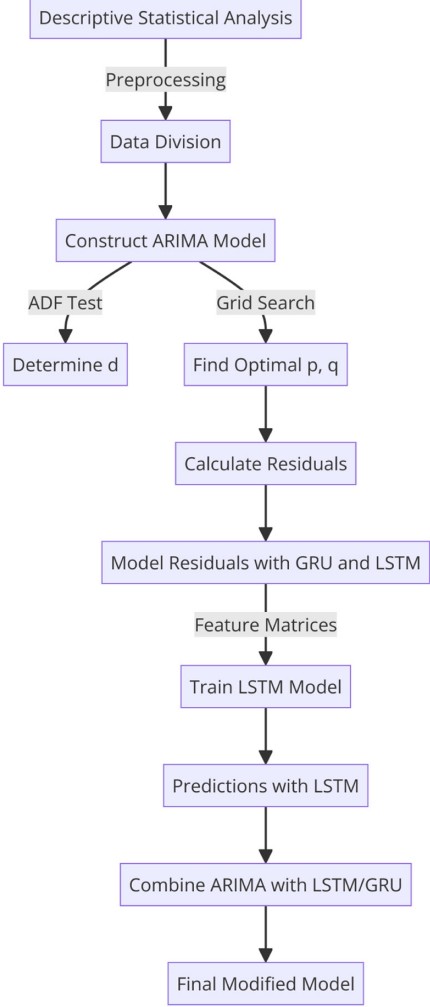

**Fig 4. Algorithmic procedure overview.**

performed to eliminate the influence of scale and improve the accuracy of predictions. Additionally, the dataset is divided into corresponding training and validation sets.

2. Construct an ARIMA model based on the preprocessed data. Initially, the order of differencing $d$ is determined through an augmented Dickey-Fuller (ADF) test, and then the values of $p$ and $q$ are determined by the process involves iteration and searching through various possible values to find the optimal parameters. In this process, an ARIMA model is constructed for each combination of $[0, p]$ and $[0, q]$, and a grid search is conducted until the Akaike information criterion (AIC) reaches its minimum value. The AIC is a widely used model selection criterion that takes into account both the goodness of fit and the complexity of the model. Ultimately, the values of $p$ and $q$ that minimize the AIC are chosen as the parameters, serving as the benchmark for determining the best model parameters [21].

3. Calculate the residuals of the optimized ARIMA model and model them using GRU and LSTM models. Taking ARIMA-LSTM as an example, the process involves constructing feature matrices for the training and testing sets based on the time series data. The training set

includes historical observations and lagged terms as input features, while the testing set only includes lagged terms as input features. Subsequently, the feature matrices of the training and testing sets are appropriately formatted to meet the input requirements of the LSTM model. A LSTM model with 32 neurons is then built, with a fully connected layer added to the model. The mean absolute error (MAE) is chosen as the loss function for the model, and the Adam optimizer is used for parameter optimization. During the training phase, the model is trained through multiple iterations using the training set, with 50 epochs and a batch size of 16. Finally, the trained model is used to make predictions on the testing set, resulting in the predicted outcomes. This approach represents the concept of a "hybrid model," where the strengths of both statistical models and machine learning models are combined. The ARIMA model captures the autocorrelation of the data, while the GRU and LSTM models are used to simulate and predict complex nonlinear relationships [23, 24].

4. Lastly, the GRU and LSTM models trained on the residuals are combined with the optimized ARIMA model to obtain the final modified model. This approach represents a typical ensemble learning method, which aims to combine the predictions of multiple models to achieve better prediction performance. The advantage of this approach is that it leverages the strengths of multiple models and can reduce bias and variance, thereby improving the accuracy and robustness of predictions [22].

## 6 Evaluation criteria

In evaluating the performance of prediction models, several evaluation criteria are commonly used. The following are some of the evaluation metrics commonly employed. In the formulas below, $f_i$ represents the predicted value from the $i$th model, $y_i$ represents the actual value, $n$ represents the number of sample data points, and $L$ denotes the maximum likelihood function value of the model:

- The Akaike Information Criterion (AIC) is a criterion used for model selection [25]. It is calculated as follows:

$$\text{AIC} = 2k - 2\ln L(\hat{\Theta}); \tag{3}$$

where $k$ is the number of parameters in the model. A lower AIC indicates a better fit or a lower complexity of the model. Therefore, the goal of model selection is to find the model that minimizes the AIC value.

- The Mean Squared Error (MSE) is a commonly used metric for assessing the accuracy of prediction models. It measures the average of the squared differences between predicted values and actual observed values. A smaller MSE value indicates better predictive performance of the model, as it implies a smaller gap between the predicted values and the actual observed values. Conversely, a larger MSE value indicates poorer predictive performance of the model.

$$\text{MSE} = \frac{1}{n}\sum_{i=1}^{n}(f_i - y_i)^2 \tag{4}$$

- Mean Absolute Error (MAE) is a commonly used metric for evaluating the accuracy of a prediction model. It represents the average of the absolute differences between the predicted values and the actual observations. Compared to Mean Squared Error (MSE), MAE does not heavily penalize large errors in the model's predictions since it does not involve squaring the differences. Therefore, MAE is equally sensitive to both large and small errors in the model's

predictions, making it more reflective of the actual prediction performance of the model, especially in the presence of noisy data.

$$\text{MAE} = \frac{1}{n}\sum_{i=1}^{n}|f_i - y_i| \tag{5}$$

- The Root Mean Squared Error (RMSE) is a commonly used metric for measuring the accuracy of prediction models. It represents the square root of the average of the squared differences between the predicted values and the actual observed values.

$$\text{RMSE} = \sqrt{MSE} \tag{6}$$

- The Mean Absolute Percentage Error (MAPE) calculates the average of the absolute differences between the predicted values and the actual values, divided by the actual values, and multiplied by 100. MAPE has an important limitation, which is that it can produce extremely large errors or undefined results when the actual values are close to or equal to zero, as dividing by zero is not possible. Therefore, MAPE is not suitable for all situations, especially when there are actual values that are zero or close to zero. In such cases, alternative error metrics may be needed.

$$\text{MAPE} = \frac{100}{n}\sum_{i=1}^{n}\left|\frac{f_i - y_i}{y_i}\right| \tag{7}$$

- The Root Mean Square Percentage Error (RMSPE) is a metric used to measure prediction errors, quantifying the root mean square of the relative errors between predicted and actual values. Similar to Root Mean Squared Error (RMSE), RMSPE also encounters a similar issue as MAPE when the actual values approach or equal zero, which may result in significant errors or undefined results.

$$\text{RMSPE} = \sqrt{\frac{1}{n}\sum_{i=1}^{n}\left(1 - \frac{y_i}{f_i}\right)^2} \tag{8}$$

- The Symmetric Mean Absolute Percentage Error (SMAPE) is a metric that differs from the traditional Mean Absolute Percentage Error (MAPE) in its symmetrical nature. SMAPE assigns equal penalty to cases where the predicted values are higher or lower than the actual values. However, it is important to note that while SMAPE provides fairness in handling overestimation and underestimation of the predictions compared to the actual values, it can become highly sensitive when the actual values approach zero.

$$\text{SMAPE} = \frac{100}{n}\sum_{i=1}^{n}\frac{|f_i - y_i|}{(|y_i| + |f_i|)/2} \tag{9}$$

## 7 Results and discussion

The performance metrics are presented in the following Table 3:

According to the MSE metric, the ARIMA-GRU model exhibits the lowest prediction error, indicating that this model has a smaller average error in forecasting the 'opening price

**Table 3. Model performance metrics.**

| Index | ARIMA | ARIMA-LSTM | LSTM | GRU | ARIMA-GRU |
|---|---|---|---|---|---|
| MSE | 0.462470 | 0.399819 | 0.4205262 | 0.426561 | **0.399764** |
| RMSE | 0.680052 | 0.632312 | 0.64848 | 0.653116 | **0.632269** |
| MAE | 0.383785 | 0.346311 | 0.3621097 | 0.367069 | **0.346333** |
| MAPE | Inf | Inf | Inf | Inf | Inf |
| SMAPE | 0.352197 | **0.334971** | 0.37839344 | 0.376948 | 0.335221 |
| RMSPE | Inf | Inf | Inf | Inf | Inf |

Note: Inf indicates cases where the MAPE and RMSPE calculations were not feasible due to division by near-zero values. Bold values represent the best performance metrics across the models.

difference' compared to other models. Therefore, based on this metric, the ARIMA-GRU model can be considered as the optimal model.

Considering the RMSE metric, the ARIMA-GRU model has the lowest standard deviation of prediction errors, which is 0.632269. Therefore, based on this metric, the ARIMA-GRU model is considered to have the best predictive performance, with relatively stable predictions that closely align with the actual observed values.

Using the Mean Absolute Error (MAE) metric to measure the average magnitude of prediction errors, we observe that the ARIMA model has an MAE of 0.383785, ARIMA-LSTM model has 0.346311, LSTM model has 0.3621097, GRU model has 0.367069, and ARIMA--GRU model has 0.346333. Based on the MAE metric, the ARIMA-LSTM model has the smallest average absolute error, indicating that its predictions have a smaller average deviation from the actual observed values. Therefore, it is considered the optimal model based on this metric.

Using the Symmetric Mean Absolute Percentage Error (SMAPE) metric to measure the relative magnitude of prediction errors, we observe that the ARIMA model has an SMAPE value of 0.352197, ARIMA-LSTM model has 0.334971, LSTM model has 0.37839344, GRU model has 0.376948, and ARIMA-GRU model has 0.335221. According to the SMAPE metric, the ARIMA-LSTM model has a relatively lower prediction error, with an SMAPE of 0.334971, indicating smaller relative errors between its predictions and the actual observed values. Therefore, based on this metric, the ARIMA-LSTM model is considered to have the best predictive performance.

## 8 Conclusion

In conclusion, different conclusions can be drawn based on different evaluation metrics. The ARIMA-GRU model exhibits superior predictive performance in terms of MSE and RMSE metrics, while the ARIMA-LSTM model performs better in terms of MAE and SMAPE metrics. The hybrid models, ARIMA-LSTM and ARIMA-GRU, outperform the individual deep learning models (LSTM and GRU) and the time series model (ARIMA) in predicting the opening price difference dataset. This indicates that combining time series models and deep learning models can enhance predictive performance by leveraging their respective strengths. Hybrid models can better capture long-term trends and short-term fluctuations in the data while possessing stronger nonlinear modeling capabilities. Therefore, in this study, the ARIMA-LSTM and ARIMA-GRU models are considered more effective predictive models. The reasons for this phenomenon can be attributed to the following three main factors:

- Advantages of the hybrid models: The hybrid models, ARIMA-LSTM and ARIMA-GRU, combine the strengths of time series models and deep learning models, effectively leveraging

their distinct characteristics in prediction. The time series model, ARIMA, is capable of capturing trends and seasonal variations in the data, while the deep learning models, LSTM and GRU, possess powerful nonlinear modeling capabilities. By combining them, the hybrid models can better capture the long-term trends and short-term fluctuations in the data, thereby improving prediction accuracy and stability.

- Adaptability to data characteristics: In the given dataset, the opening price difference may exhibit certain nonlinear relationships and temporal dependencies. Deep learning models are better at capturing and modeling such nonlinear relationships, while time series models can account for the temporal dependencies in the data. By combining these two types of models, the hybrid models can comprehensively consider the data's characteristics, thereby enhancing prediction accuracy.

- Parameter optimization and tuning: During the model development process, we performed parameter optimization and tuning for each model to achieve optimal performance. For the ARIMA model, we employed automated parameter selection methods to ensure a good fit of the model. For the deep learning models, we optimized the network structure and adjusted hyperparameters to achieve the best predictive results.

In conclusion, the aforementioned conclusions are attributed to the ability of the hybrid models, ARIMA-LSTM and ARIMA-GRU, to effectively leverage the advantages of time series models and deep learning models, thereby capturing the trends and fluctuations in the data more accurately. Additionally, the optimization of parameters and experimental design provide support for the reliability of these conclusions. The findings of this study hold significant value for decision-making and investment strategy formulation in related fields. In addressing the weaknesses and limitations of this study, it is essential to critically assess the contribution and interpretability of the deep learning models within our hybrid modeling approach. While the integration of ARIMA with LSTM and GRU models has demonstrated enhanced predictive accuracy for the opening price spread, a notable challenge lies in the opaque nature of deep learning models, often referred to as "black boxes." This aspect can hinder our ability to fully understand and explain the specific factors driving the model's predictions. Consequently, although our hybrid models capitalize on the strengths of both time series and deep learning techniques to capture market dynamics effectively, the inherent complexity and lack of transparency in the deep learning components may limit the interpretability of the results. The future research directions include:

1. Identification of Key Covariates Influencing the Opening Price Spread: Future studies should focus on uncovering the critical covariates that significantly impact the opening price difference rate. This entails conducting comprehensive analyses to isolate and understand the effects of various economic, financial, and socio-political factors that may influence the opening price spread. Identifying these key covariates will not only enhance the understanding of the dynamics at play but also improve the predictive accuracy of models concerning opening price behavior.

2. More Effective Autoregressive Models for Predicting Opening Price Spread: Building upon the foundation laid by this study, future work should aim at innovating and refining autoregressive models specifically tailored for predicting the opening price spread. This includes exploring advanced statistical techniques, incorporating machine learning algorithms, and integrating high-frequency trading data to capture the nuances of price movements more accurately. The goal is to develop models that are not only robust and reliable but also capable of adapting to the evolving nature of financial markets.

3. Exploring the Fundamental Mechanisms Behind Opening Price Spread Formation: Beyond predictive modeling, it is imperative to investigate the underlying mechanisms that give rise to the opening price spread from a price formation perspective. This research avenue involves a detailed examination of the market microstructure, the role of investor sentiment, and the impact of overnight news and events on price setting at market open. Understanding the essence of how opening price spreads are formed will contribute significantly to the knowledge of market efficiency, liquidity dynamics, and the broader economic implications tied to these phenomena.

Through a combination of empirical investigation and theoretical exploration, we aim to uncover deeper insights into the opening price spread, thereby aiding investors, market analysts, and policymakers in making more informed decisions.

## Supporting information

**S1 Data.**
(CSV)

## Author Contributions

**Conceptualization:** Yuancheng Si, Saralees Nadarajah.

**Data curation:** Yuancheng Si.

**Formal analysis:** Yuancheng Si, Saralees Nadarajah.

**Funding acquisition:** Yuancheng Si.

**Investigation:** Yuancheng Si, Zongxin Zhang.

**Methodology:** Yuancheng Si, Saralees Nadarajah, Zongxin Zhang.

**Project administration:** Yuancheng Si, Zongxin Zhang.

**Resources:** Yuancheng Si, Chunmin Xu.

**Validation:** Chunmin Xu.

**Visualization:** Yuancheng Si, Saralees Nadarajah, Chunmin Xu.

**Writing – original draft:** Yuancheng Si.

**Writing – review & editing:** Yuancheng Si, Zongxin Zhang.

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
