## [Decision Letter · Decision Letter 0]

26 Jan 2024

PONE-D-23-36601Modeling Opening Price Spread of Shanghai Composite Index Based on ARIMA-GRU/LSTM Hybrid ModelPLOS ONE

Dear Dr. Si,

Thank you for submitting your manuscript to PLOS ONE. After careful consideration, we feel that it has merit but does not fully meet PLOS ONE’s publication criteria as it currently stands. Therefore, we invite you to submit a revised version of the manuscript that addresses the points raised during the review process.

**ACADEMIC EDITOR: **

The reviewers have provided detailed feedback that highlights several areas needing significant improvement. Below, I have summarised the key points from the reviewers, along with additional guidance on how to address these concerns effectively.

Reviewer #1 and #3 have pointed out the need for a more concise and structured abstract. Ensure that the abstract clearly states the research problem, methodology, main findings, and significance. It should not exceed 250 words.Reviewer #1 suggests incorporating a data link within the manuscript for transparency and reproducibility. Please provide a direct link or a DOI where the data can be accessed.There is inconsistency noted in section numbering. Decide on a format and apply it consistently throughout the document.Table 3 should highlight the most significant values. Consider using bold text or color-coding to draw attention to key data.Clarify the choice of the ARIMA model in the Model Composition section as recommended by Reviewer #1.Include more recent studies in your references to ensure the research is grounded in current knowledge, as suggested by Reviewer #1.Review and correct the citation indexing, starting with [1] as opposed to [13], and ensure all citations are present and correctly numbered.Reviewer #2 requests a revision of the abstract to include the introduction/significance of the study, research aims, methodology, and main conclusions.Ensure that acronyms are defined at first mention and used consistently thereafter.A section outlining the paper's organization should be included at the end of the introduction. Also, add a comprehensive literature review to demonstrate the novelty of your study.All variables used in equations should be clearly explained for the reader's understanding.Discuss the implications, limitations, and directions for future research to provide context and potential for further study.The paper currently lacks a dedicated conclusion section, which is imperative for summarising the study and its findings.Address the technical issues mentioned by Reviewer #2 and conduct a meticulous proofread to enhance the paper's quality.Reviewer #3 suggests revising the title to make it more engaging. Also, the abstract should be rewritten to focus more on your research rather than the background.Include a clear algorithm or process flow to support the discussions in Section 2, as indicated by Reviewer #3.Ensure that all tables follow the PLoS ONE format, or provide justification for the current format.Minimize the use of personal pronouns such as "Our" and replace them with an impersonal language that suits academic writing.==============================

We look forward to receiving your revised manuscript.

Kind regards,

Dr. Muhammad Usman Tariq

Academic Editor

PLOS ONE

Reviewers' comments:

Reviewer's Responses to Questions

**Comments to the Author**

1. Is the manuscript technically sound, and do the data support the conclusions?

Reviewer #1: Yes

Reviewer #2: Partly

Reviewer #3: Yes

2. Has the statistical analysis been performed appropriately and rigorously? 

Reviewer #1: Yes

Reviewer #2: Yes

Reviewer #3: Yes

3. Have the authors made all data underlying the findings in their manuscript fully available?

Reviewer #1: No

Reviewer #2: Yes

Reviewer #3: No

4. Is the manuscript presented in an intelligible fashion and written in standard English?

Reviewer #1: Yes

Reviewer #2: Yes

Reviewer #3: No

5. Review Comments to the Author

Reviewer #1: • Begin with a concise Abstract that addresses the identified issues in the forecast stock market, outlining the objectives aimed at addressing these gaps. Subsequently, provide an overview of the chosen methods, as detailed in the current Abstract. Conclude with a brief summary of the results, emphasizing their significance clearly and straightforwardly.

• It is recommended to incorporate the data link, specifying where it can be downloaded, into the manuscript.

• Please verify the formatting to determine whether each section should be numbered or not

• The author is encouraged to emphasize the highest value in Table 3 through techniques such as highlighting, color-coding, or bold formatting.

• It is suggested that the reasons for selecting ARIMA as the baseline model in Section Results and Discussion, paragraph 1, be elucidated in the Model Composition-ARIMA section.

• Regarding the references provided, the author has 10 recent publications (47%) out of 21 listed. A recent publication is counted from 2019 onwards. Thus, authors are advised to add more recent publications to support their literature.

• The citation indexing seems wrong. It should start with [1] instead of [13]. Please redo the citation.

• Please review all your citations, as the citation index for bullet point number 2 in the Introduction Section appears to be missing.

Reviewer #2: The chapter is in good shape, but it needs to be modified before resubmission. There are a few comments that may help to enhance the quality of the paper.

1. An abstract need to be revised and expanded: Abstract of a research paper is typically 200 to 400 words in length, and 150 to 300 words for a review paper.

2. The abstract should highlight the objectives of the contribution. Remember, an abstract is often the first and sometimes the only part of a document that people read, so it should effectively convey the main points and encourage further exploration.

3. The abstract is wordy and not informative. The structure of the abstract needs revision. Revise the abstract to provide.

a. the introduction/significance of the study,

b. the aim of the study,

c. the research methodology,

d. the major conclusion of the study

4. To ensure clarity and consistency, Gated Recurrent Unit (GRU) is spelled out initially, and then "GRU" is used as the abbreviation throughout the rest of the chapter.

5. The organization of the paper is missing that needs to be added at end of the introduction section after the main contributions.

6. The related work (literature review) part is missing. It should be added and discuss the current result with previous studies to show the novelty of the study.

7. Related work or literature review can be highlighted. However, it is difficult to compare the current study with previous studies.

8. The author should explain all the variables used in the equation.

9. implications, limitations and future work can be included.

10. The paper lacks a dedicated conclusion section, which is an essential component of scholarly writing.

11. The manuscript should be read more carefully to improve the paper quality because some technical weaknesses are found in it.

Reviewer #3: A. Main title is not attractive – the author should change the main title of the article.

B. Authors should follow the following procedure for abstract writing

Abstract is vague: The abstract needs to be rewritten professionally; there should be some background knowledge of the area, existing problems, and novel ideas that address the problem.

a) Firstly, an abstract should summarize the major aspects of the entire paper:

b) The overall purpose of the study

c) Basic methodology of your research

d) Major findings as a result of your analysis

e) A brief summary of your interpretations and conclusions.

Please adjust your Abstract according to the aforementioned logic and ensure that most of your Abstract is about your research, not the context. Most of your abstract states the inappropriate research background. Please add more content about your research. What's more, an abstract of a research paper is typically 150 to 250 words; please modify it. Moreover, it is suggested that remove the personal pronoun "Our", which is found extensively throughout the paper, and replace it with something like "This paper..." or "This research work...".

C. Contribution statements are vague

D. Motivation is missing

E. The figure 3 doesn’t elaborate the main idea

F. In section 2, algorithm is not given. The author must demonstrate the algorithm for which they expressed their views.

G. Weather the tables are according to PLoS ONE format, if not, then tell us what you meant by table 3

H. Separate conclusion is required.

6. PLOS authors have the option to publish the peer review history of their article (what does this mean?). If published, this will include your full peer review and any attached files.

Reviewer #1: **Yes: **ASRAFUL SYIFAA AHMAD

Reviewer #2: **Yes: **Samina Amin

Reviewer #3: No

---

## [Author Response · Author response to Decision Letter 0]

1 Feb 2024

Response to Reviewers

Reviewer #1: • Begin with a concise Abstract that addresses the identified issues in the forecast stock market, outlining the objectives aimed at addressing these gaps. Subsequently, provide an overview of the chosen methods, as detailed in the current Abstract. Conclude with a brief summary of the results, emphasizing their significance clearly and straightforwardly.

Response to Reviewer #1:We sincerely appreciate your valuable and constructive feedback on our manuscript. Your insightful suggestions have been instrumental in guiding us to substantially improve the quality of our work. We have carefully considered each of your recommendations and have made corresponding revisions to our manuscript, particularly focusing on enhancing the clarity, depth, and rigor of our abstract.

• It is recommended to incorporate the data link, specifying where it can be downloaded, into the manuscript.

Response to Reviewer #1:Thank you very much for your valuable suggestion regarding the inclusion of a data link in our manuscript. We deeply appreciate your attention to detail and your commitment to ensuring the accessibility and reproducibility of our research.

In response to your recommendation, we have updated the manuscript to include specific information on where the data can be downloaded. We have provided the website address where the dataset for the Shanghai Composite Index, covering the period from December 20, 1990, to June 2, 2023, is publicly available.

Please note that accessing the dataset requires registration on the website or contacting the corresponding author via email to request the data. We believe this procedure ensures the data's integrity while making it accessible to researchers and practitioners interested in replicating or extending our study.

We have made the necessary modifications to the manuscript to reflect this change and hope it meets your approval. Once again, we are grateful for your insightful comments and guidance, which have undeniably enhanced the quality of our work.

• Please verify the formatting to determine whether each section should be numbered or not

Response to Reviewer #1:Thank you very much for your guidance regarding the formatting of our manuscript, specifically the numbering of each section. 

Following your suggestion, we have carefully reviewed the formatting guidelines of the journal and adjusted the manuscript accordingly. Each section is now numbered consistently throughout the document to ensure clarity and ease of navigation for readers.

• The author is encouraged to emphasize the highest value in Table 3 through techniques such as highlighting, color-coding, or bold formatting.

Response to Reviewer #1:Thank you very much for your constructive suggestion regarding the presentation of Table 3 in our manuscript. Your advice to emphasize the highest value in each performance metric through highlighting, color-coding, or bold formatting is greatly appreciated and has been instrumental in enhancing the clarity and impact of our findings.

In response to your recommendation, we have revised Table 3 by marking the most significant model figures for each indicator in bold red. This adjustment not only makes it easier for readers to identify the key results at a glance but also accentuates the superior performance of certain models in our analysis.

• It is suggested that the reasons for selecting ARIMA as the baseline model in Section Results and Discussion, paragraph 1, be elucidated in the Model Composition-ARIMA section.

Response to Reviewer #1: Thank you for your valuable suggestion. We recognize the importance of providing a comprehensive rationale for our methodological choices to our readers and appreciate your guidance in this regard.

In response to your recommendation, we have revised the manuscript structure accordingly. The "Model Composition-ARIMA" section now includes a detailed explanation of our decision to utilize ARIMA as the baseline model for our analysis. This amendment ensures a clearer understanding of the model's significance and its foundational role in our research.

• Regarding the references provided, the author has 10 recent publications (47%) out of 21 listed. A recent publication is counted from 2019 onwards. Thus, authors are advised to add more recent publications to support their literature.

Response to Reviewer #1: Thank you sincerely for your valuable suggestion regarding the inclusion of more recent publications in our references to strengthen and update our literature review. We deeply appreciate your attention to the relevance and timeliness of the research we cite, which is crucial for maintaining the academic rigor and currency of our work. In response to your advice, we have carefully reviewed the latest literature and are pleased to inform you that we have incorporated an additional three recent publications into our reference list. Notably, two of these publications focus on the analysis of the Taiwan stock market, with both being published after September 2023. These recent studies provide cutting-edge insights into the dynamics of the Taiwan stock market, further enriching our discussion and supporting our analysis with the most current research findings.

We are grateful for your guidance, which has significantly contributed to enhancing the quality and relevance of our manuscript. Your suggestion has helped us ensure that our literature review reflects the latest developments and scholarly discussions in the field.

• The citation indexing seems wrong. It should start with [1] instead of [13]. Please redo the citation.

• Please review all your citations, as the citation index for bullet point number 2 in the Introduction Section appears to be missing.

Response to Reviewer #1:Thank you for bringing the issues regarding citation indexing and the missing citation in bullet point number 2 of the Introduction Section to our attention. Your meticulous review of our manuscript and detailed feedback are truly invaluable to us.

Following your suggestions, we have thoroughly reviewed and corrected the citation indexing throughout our manuscript, ensuring that it now starts with [1] as it should. Additionally, we have addressed the missing citation in the Introduction Section, ensuring that all statements are appropriately supported by relevant references.

Reviewer #2: The chapter is in good shape, but it needs to be modified before resubmission. There are a few comments that may help to enhance the quality of the paper.

1. An abstract need to be revised and expanded: Abstract of a research paper is typically 200 to 400 words in length, and 150 to 300 words for a review paper.

2. The abstract should highlight the objectives of the contribution. Remember, an abstract is often the first and sometimes the only part of a document that people read, so it should effectively convey the main points and encourage further exploration.

3. The abstract is wordy and not informative. The structure of the abstract needs revision. Revise the abstract to provide.

a. the introduction/significance of the study,

b. the aim of the study,

c. the research methodology,

d. the major conclusion of the study

Response to Reviewer #2:Thank you immensely for your constructive comments and suggestions aimed at enhancing the quality of our paper. We are truly appreciative of the time and effort you have devoted to reviewing our work and providing such detailed feedback.

In response to your valuable guidance, we have undertaken a thorough revision of our abstract. Acknowledging your observation that our initial abstract was wordy and lacked clarity, we have expanded and restructured it to ensure it falls within the recommended length of 200 to 400 words for a research paper. 

4. To ensure clarity and consistency, Gated Recurrent Unit (GRU) is spelled out initially, and then "GRU" is used as the abbreviation throughout the rest of the chapter.

Response to Reviewer #2:We are grateful for your suggestion regarding the use of abbreviations in our chapter, particularly your advice on the consistent use of "Gated Recurrent Unit (GRU)" and its abbreviation. Your attention to detail and emphasis on clarity and consistency are highly valued and have significantly contributed to enhancing the readability and professionalism of our manuscript.

In line with your recommendation, we have carefully revised our chapter to ensure that "Gated Recurrent Unit (GRU)" is fully spelled out at its first mention, with the abbreviation "GRU" consistently used thereafter. Similarly, we have applied this approach to other key terms within our chapter, such as "Autoregressive Integrated Moving Average (ARIMA)" and "Long Short-Term Memory (LSTM)", to maintain uniformity and clarity throughout the text.

5. The organization of the paper is missing that needs to be added at end of the introduction section after the main contributions.

Response to Reviewer #2: We are grateful for your suggestion, in response to the valuable feedback provided by the reviewer, we have carefully revised the manuscript to include a detailed outline of the paper's organization at the end of the introduction section. This addition aims to enhance the clarity and navigability of the paper for our readers. We sincerely appreciate the reviewer's constructive suggestions and have endeavored to address them thoroughly in our revision. Thank you for the opportunity to improve our work.

6. The related work (literature review) part is missing. It should be added and discuss the current result with previous studies to show the novelty of the study.

Response to Reviewer #2: We sincerely appreciate the reviewer's insightful suggestions and have incorporated a detailed literature review section to address this feedback. This addition not only contextualizes our research within the existing body of knowledge but also highlights the novel contributions of our study. We are grateful for the guidance provided, which has undoubtedly strengthened the quality and depth of our manuscript.

7. Related work or literature review can be highlighted. However, it is difficult to compare the current study with previous studies.

Response to Reviewer #2: We deeply appreciate the reviewer's observations and understand the concern regarding the comparison of our study with previous research. Our investigation targets a relatively new area within financial modeling, employing novel modeling techniques that diverge from traditional approaches. Consequently, the scarcity of literature reviews directly related to our specific research domain is notable. Most existing studies in this area have primarily focused on exploring the underlying causes of opening price differences, which does not align directly with our research direction, emphasizing predictive modeling using advanced techniques. We acknowledge this gap and are actively seeking to bridge it with our current and forthcoming research. We have similar works under review and would greatly value any insightful comments and suggestions from the reviewer in the future. This feedback will be instrumental in refining our research and contributing meaningfully to the field.

8. The author should explain all the variables used in the equation.

Response to Reviewer #2:We sincerely appreciate the reviewer's constructive feedback regarding the explanation of variables within our equations. Following your suggestion, we have meticulously reviewed our manuscript and ensured that each variable used in our equations is now clearly defined and explained. This enhancement aims to improve the clarity and comprehensibility of our mathematical modeling, facilitating a better understanding of our research methodology for readers. We are grateful for this opportunity to refine our work and thank the reviewer for their valuable input.

9. implications, limitations and future work can be included.

Response to Reviewer #2: In response to the reviewer's valuable feedback, we have meticulously incorporated sections on implications, limitations, and future work into the conclusion of our manuscript. We express our sincere gratitude for the constructive suggestions provided, as they have significantly enriched the depth and scope of our study. Through these additions, we aim to not only highlight the practical relevance of our findings but also acknowledge the constraints of our research approach and outline promising avenues for subsequent investigations. We hope these revisions meet the reviewer's expectations and contribute to a more comprehensive understanding of our study's contributions to the field.

10. The paper lacks a dedicated conclusion section, which is an essential component of scholarly writing.

Response to Reviewer #2:In response to the reviewer's insightful observation, we have now added a dedicated conclusion section to our manuscript. We appreciate the guidance provided, recognizing the importance of a conclusion in encapsulating the key findings, implications, and future directions of our research. This addition aims to succinctly summarize the study's contributions to the field, offering readers a clear understanding of its significance and potential impact. We are grateful for the opportunity to enhance our manuscript based on the reviewer's valuable feedback.

11. The manuscript should be read more carefully to improve the paper quality because some technical weaknesses are found in it.

Response to Reviewer #2:In response to the reviewer's comment, we have conducted a thorough review and revision of our manuscript to address the technical weaknesses identified. We acknowledge that there may still be areas for improvement and sincerely appreciate the feedback provided. We commit to continuous efforts in refining our work and enhancing the quality of our paper. Thank you for bringing these issues to our attention, and we welcome any further suggestions that can aid in our manuscript's improvement.

Reviewer #3: A. Main title is not attractive – the author should change the main title of the article.

Response to Reviewer #3: We sincerely appreciate your valuable suggestion to make the main title of our article more attractive. Currently, our manuscript is undergoing a major revision process. We are uncertain about the PLOS ONE policy regarding title changes at this stage of the revision. Therefore, we have not modified the title as of now. However, we are open to reconsidering and adjusting the title to better reflect the content and appeal of our research, should the opportunity arise during the later stages of the review process or as permitted by the journal's guidelines.

B. Authors should follow the following procedure for abstract writing

Abstract is vague: The abstract needs to be rewritten professionally; there should be some background knowledge of the area, existing problems, and novel ideas that address the problem.

a) Firstly, an abstract should summarize the major aspects of the entire paper:

b) The overall purpose of the study

c) Basic methodology of your research

d) Major findings as a result of your analysis

e) A brief summary of your interpretations and conclusions.

Please adjust your Abstract according to the aforementioned logic and ensure that most of your Abstract is about your research, not the context. Most of your abstract states the inappropriate research background. Please add more content about your research. What's more, an abstract of a research paper is typically 150 to 250 words; please modify it. Moreover, it is suggested that remove the personal pronoun "Our", which is found extensively throughout the paper, and replace it with something like "This paper..." or "This research work...".

Response to Reviewer #3: We sincerely thank you for your constructive comments and have revised our abstract accordingly. We have ensured that the abstract now succinctly summarizes the major aspects of our paper, including the study's purpose, the basic methodology employed, the major findings from our analysis, and a brief summary of our interpretations and conclusions. We have also adjusted the length of the abstract to fit within the recommended range of 150 to 250 words and carefully replaced personal pronouns such as "Our" with "This paper" or "This research work" to maintain a professional and objective tone throughout the manuscript. We believe these revisions have significantly improved the clarity and professionalism of the abstract, and we are grateful for the guidance provided.

C. Contribution statements are vague

Response to Reviewer #3: In response to the reviewer's feedback regarding the vagueness of our contribution statements, we have taken your suggestions into careful consideration and have made the necessary revisions. We have now explicitly outlined our contribution statements in Section 3 of the paper, ensuring they are clearly defined and articulated. This adjustment aims to highlight the unique contributions of our research more effectively and to provide readers with a concise understanding of the value and novelty of our work. We are grateful for your insightful recommendations and believe that these changes significantly enhance the clarity and impact of our manuscript.

D. Motivation is missing

Response to Reviewer #3:In response to Reviewer #3's feedback regarding the lack of explicit motivation in our manuscript, we express our sincere appreciation for bringing this to our attention. We have now included a section dedicated to elucidating the motivation behind our study. This section highlights the significance of exploring the opening price spread of the Shanghai Composite Index and the potential impact of our findings on financial modeling and forecasting practices. By addressing the challenges associated with traditional and contemporary modeling techniques, we aim to underscore the necessity and relevance of our research approach.

E. The figure 3 doesn’t elaborate the main idea

Response to Reviewer #3:

Many thanks for your suggestion, In response to your feedback, we have carefully considered the role of Figure 3 within our chapter and have decided to remove it.

F. In section 2, algorithm is not given. The author must demonstrate the algorithm for which they expressed their views.

Response to Reviewer #3: We greatly appreciate your valuable feedback on our manuscript. Following your suggestion, we have now included a comprehensive flowchart to visually depict the algorithm's workflow. Additionally, we have elaborated on the specifics of the algorithm within the text, ensuring a detailed explanation of its functionality and application in our research. These enhancements are aimed at providing a clearer understanding of the algorithmic approach we adopted. We hope that these revisions will adequately address your concerns and improve the manuscript's clarity and depth of technical detail.

G. Weather the tables are according to PLoS ONE format, if not, then tell us what you meant by table 3

Response to Reviewer #3: We greatly appreciate your attention to detail and guidance regarding the formatting of our tables. Following your recommendation, we have meticulously revised all tables, including Table 3, to ensure full compliance with the PLOS ONE formatting guidelines. This includes adjustments to table structure, captioning, and the addition of necessary legends and footnotes for clarity. We hope these revisions adequately address your concerns and enhance the presentation and readability of our data in alignment with the journal's standards.

H. Separate conclusion is required.

Response to Reviewer #3:

In response to the reviewer's feedback, we sincerely appreciate the guidance provided and have accordingly incorporated a dedicated conclusion section into our manuscript and we are grateful for the opportunity to improve our work based on your valuable feedback.

We express our deepest gratitude to all reviewers for their meticulous review and insightful comments on our manuscript. Your detailed feedback has been invaluable in guiding the enhancements and revisions of our work. We have taken each suggestion and critique into careful consideration, making dedicated efforts to address the identified issues and improve the overall quality, clarity, and impact of our research. The constructive feedback provided by the reviewers has undeniably enriched our manuscript, making it a more comprehensive and robust contribution to the field. We hold immense respect for the review process and sincerely appreciate the time and expertise that the reviewers have contributed to refining our work. Thank you for your support and guidance, which have been instrumental in elevating the quality of our paper.

---

## [Editor Report · Decision Letter 1]

6 Feb 2024

Modeling Opening Price Spread of Shanghai Composite Index Based on ARIMA-GRU/LSTM Hybrid Model

PONE-D-23-36601R1

Dear Dr. Si,

We’re pleased to inform you that your manuscript has been judged scientifically suitable for publication and will be formally accepted for publication once it meets all outstanding technical requirements.

Kind regards,

Muhammad Usman Tariq

Academic Editor

PLOS ONE
---

## [Editor Report · Acceptance letter]

27 Feb 2024

PONE-D-23-36601R1 

PLOS ONE

Dear Dr. Si, 

I'm pleased to inform you that your manuscript has been deemed suitable for publication in PLOS ONE. Congratulations! Your manuscript is now being handed over to our production team.

Kind regards, 

on behalf of

Dr. Muhammad Usman Tariq 

Academic Editor

PLOS ONE